# Lactic Acid Modified Natural Rubber–Bacterial Cellulose Composites

**Sirilak Phomrak and Muenduen Phisalaphong \***

Department of Chemical Engineering, Faculty of Engineering, Chulalongkorn University,
Bangkok 10330, Thailand; loogpoo@hotmail.com
\* Correspondence: muenduen.p@chula.ac.th; Tel.: +662-218-6875

**Abstract:** Green composite films of natural rubber/bacterial cellulose composites (NRBC) were prepared via a latex aqueous microdispersion process. The acid modified natural rubber/bacterial cellulose composites (ANRBC), in which lactic acid was used, showed significant improvement in mechanical properties, melting temperature, and high resistance to polar and non-polar solvents. The ANRBC films exhibited improved water resistance over that of BC and NRBC films, and possessed a higher resistance to non-polar solvents, such as toluene, than NR and NRBC films. The modification had a slight effect on the degradability of the composite films in soil. The NRBC and ANRBC films were biodegradable; the NRBC80 and ANRBC80 films were degraded completely within 3 months in soil. NRBC and ANRBC showed no antibacterial activity against *Escherichia coli* and *Staphylococcus aureus* and did not show cytotoxic effects on the HEK293 and HaCaT cell lines.

**Keywords:** bacterial cellulose; natural rubber; composites; biomaterials; biopolymers

---

## 1. Introduction

Natural rubber (NR) is a raw material for a variety of elastomer industries, such as tires, gloves, automotive interior parts and packaging, because of its excellent elastic properties. However, NR has some less desirable properties, such as low strength and abrasion resistance. NR's mechanical properties vary with temperature; the softness of NR increases with increasing temperature, whereas its brittleness increases at low temperatures. NR has poor chemical resistance to non-polar solvents such as acetone and benzene. Reinforcement is used as a method for increasing NR's mechanical properties, for instance its modulus and strength. Carbon black and silica nanoparticles have been widely used as reinforcing agents [1–3]. Bacterial cellulose (BC) is a nanocellulose that has been studied for use in reinforcing NR [4]. BC is produced by *Acetobacter xylinium* in the form of nanofibers. Eventually, the nanofibrils bundle to form microbial cellulose ribbons. BC fibers have a significantly smaller diameter compared with plant fibers. Specifically, BC has a more crystalline structure and has a short-time synthesis process when compared with plant celluloses [5–7]. The main obstruction to reinforcing NR with BC is that the two materials are incompatible because of their polarity differences. NR is a non-polar material, whereas BC is a polar material.

Cellulose has a strong affinity toward materials containing hydroxyls groups. Cellulose swells in a high-polarity solvent and its internal hydrogen bonds weaken in the presence of a large number of hydroxyl groups. Previously, for the fabrication of a BC and NR composite, BC was treated to decrease its polarity and NR was grafted with high polarity functional groups on its structure. The composite fabrication via the solution blending method also helped reduce the hydrophilicity and polarity of the nanofibers; hence, this method improves the adhesion of cellulose fibers to the NR matrix [4,7]. The addition of BC into the NR matrix affects the properties of the NR composites, such as reduced toughness and increased polymer swelling in the polar solvents [4]. In addition, crosslinking has

been applied as a method for improving properties of rubber. The network structure formed under crosslinking conditions is a key leading to modified properties. One of the earliest examples of the crosslinking process is the vulcanization of rubber by adding sulfur under high temperature, which creates links between the latex molecules [8]. Vulcanization gives the rubber its strength over temperature ranges in which non-vulcanized rubber could not perform. The cross links increase the toughness, strength, and hardness of rubber composites. In particular, cross-linked rubber is resistant to attack from oxidizing agents. However, because of the side-effects of sulfur, such as undesirable colors and toxicity, organic acids (e.g., formic, maleic, stearic, citric, and dicarboxylic acid) have been applied and evaluated in cross-linking studies of NR composites [9–12].

Previously, NRBC composite films had been successfully prepared via a latex aqueous microdispersion process by thoroughly mixing BC slurry with natural rubber latex (NRL) [4]. In this work, lactic acid cross-linking was applied to improve the properties of NRBC composites. The goal of this research is to develop biodegradable films of NRBC composites with improved mechanical properties, good resistance in water and non-polar solvents, together with high thermal stability and non-toxicity in order to be further applied in the fields of packaging and biomaterials.

## 2. Materials and Methods

### 2.1. Materials

NRL with 60 wt.% dry rubber content (DRC) was purchased from the Rubber Research Institute of Thailand (Bangkok, Thailand). BC (≈99% water content in wet weight) was kindly provided by Pramote Thamarat from the Institute of Research and Development of Food Product, Kasetsart University (Bangkok, Thailand). All other chemical reagents were purchased from Sigma-Aldrich (Thailand) Co., Ltd. (Bangkok, Thailand).

### 2.2. Particle Size Analysis

Particle size analysis of BC fibers in DI water and NR particles in NRL was performed by dynamic light scattering (DLS) using a Malvern Zetasizer Nano-ZS Version 7.04 (Malvern Instruments, Malvern, UK). Particle size and size distribution (PDI) were measured at 25 °C.

### 2.3. Preparation of NRBC Composites

For the preparation of natural rubber (NR) film, NRL had to be diluted into 30 wt.% dry rubber content before fabrication. For the preparation of NRBC films, the dilute slurry of BC (1 wt.% in DI water) was added to the diluted NRL at various weight ratios of NR/BC (dry basis) at 80/20 (NRBC20), 50/50 (NRBC50), and 20/80 (NRBC80); for example, to prepare the mixture of NRBC20 for 100 g, 11.8 g of 30 wt.% NRL was mixed with 88.2 g of 1 wt.% BC slurry. The mixture was thoroughly mixed using a mechanical stirrer (200 rpm) for 5 min at room temperature. The films were fabricated by pouring the prepared mixture of 100 g into a stainless-steel tray, setting at room temperature for 3 h, and drying in an air convection oven at 50 °C for 2 days.

### 2.4. Modification with Lactic Acid

Dried films of NRBCs were immersed in 400 mL of 0.25 M lactic acid for 2 h at room temperature (30 °C). Then the films were washed with DI water and dried in an air convection oven at 50 °C for 24 h. ANRBC represents the lactic acid modified natural rubber/bacterial cellulose composite. The area and thickness of the prepared films are shown in Table 1.

**Table 1.** Area and thickness of films.

| Samples | Area ($cm^2$) | Thickness (mm) |
| --- | --- | --- |
| NR | 400 | 0.27 ± 0.02 |
| NRBC20 | 400 | 0.17 ± 0.02 |
| ANRBC20 | 400 | 0.18 ± 0.02 |
| NRBC50 | 400 | 0.06 ± 0.01 |
| ANRBC50 | 400 | 0.06 ± 0.01 |
| NRBC80 | 400 | 0.03 ± 0.01 |
| ANRBC80 | 400 | 0.03 ± 0.01 |
| BC | 400 | 0.04 ± 0.00 |

*2.5. Characterization of Composite Films*

The cross-section morphology of composite films was examined by scanning electron microscopy (SEM). The images were immediately viewed at an accelerating voltage of 15 kV under SEM using a JEOL JSM-5410LV microscope (Tokyo, Japan) [4].

Attenuated total reflectance Fourier transform infrared spectroscopy (ATR–FTIR) spectra of the films were measured at wavenumbers ranging from 4000 to 600 $cm^{-1}$ at a resolution of 4 $cm^{-1}$ with a Nicolet SX-170 FTIR spectrometer (Thermo Scientific, Waltham, MA, USA).

The static water contact angles (1 μL drop size) were measured by using the sessile drop method. The images of the drop were captured and evaluated by using a Kruss Drop Shape Analyzer (DSA 10 Mk2).

Crystallinities of films were characterized by an X-ray diffractometer (model D8 Discover, Bruker AXS, Karlsruhe, Germany). X-ray diffraction patterns were recorded with CuKa radiation (k = 1.54 Å). The operating voltage and current were 40 kV and 40 mA, respectively. Samples were scanned from 10 to 40° 2θ at a scan speed of 3° $min^{-1}$. Profile fitting and crystallinity (%) calculations were performed with Topas version 3.0 (Bruker, AXS) software.

Thermogravimetric analysis (TGA) was performed on a TGA Q50 V6.7 Build 203, Universal V4.5A (TA Instruments, New Castle, DE, USA) equipped with a platinum cell. The scanning range was 30–600 °C using a heating rate of 10 °C $min^{-1}$. The temperature at maximum weight loss rate ($T_{max}$) was measured from the DTG curves. The melting temperature ($T_m$) and glass transition temperature ($T_g$) were measured by differential scanning calorimetry (DSC). Samples about 3–5 mg were sealed in an aluminum pan. Samples were measured under a nitrogen atmosphere. Nonisothermal DSC analysis of samples was performed using a NETZSCH DSC 204 F1 Phoenix (Selb, Germany). The samples were heated from −100 to 300 °C at a rate of 10 °C $min^{-1}$ using a cool–heat mode.

Mechanical properties were measured in terms of Young's modulus, tensile strength, and elongation at break. Films were cut into strips 1 cm wide and 10 cm long. The mechanical properties of films were determined with a Lloyd 2000R (Southampton, UK) universal testing machine. The test conditions followed ASTM D882. The Young's modulus, tensile strength, and elongation at break were the average values determined from five specimens.

Films were determined for toluene and water uptake at room temperature according to ASTM D471. Each test specimens were in the form of $2 \times 2$ $cm^2$. The weight of each samples were measured before immersion into toluene and water for 1, 2, 3, and 4 weeks. The degree of swelling was calculated as

$$\text{Degree of swelling}(\%) = \frac{W_w - W_d}{W_d} \times 100 \qquad (1)$$

where $W_w$ and $W_d$ represent the weight of wet and dry films, respectively. The average values were determined from three specimens.

Water vapor transmission rates (WVTRs) of the NRBC and ANRBC dry films were determined with a water vapor permeation tester (Labthink Model W3/031). The test conditions followed ASTM E 96-00 water vapor transmission of material. WVTR testing was analyzed under 38 °C and 90% relative

humidity. The principle of this measurement was similar to that of the conventional method in that one side of the film was exposed to water vapor. As water solubilized into the film and permeated through the sample material, nitrogen gas swept and transported the transmitted water vapor molecules to a calibrated infrared sensor on the other side. Thereafter, the transmission rate was determined.

Biodegradation of NRBC and ANRBC composites in soil was carried out for 6 months. Each test specimen was in the form of $5 \times 5$ cm$^2$ and buried to a 10 cm soil depth in a pot under the natural soil environment, where the soil moisture was around 11–14 (% vol) and soil temperature was around 27–33 °C. The samples were weighed after 1, 3, and 6 months. For calculation of biodegradation, the samples were carefully taken out, washed with distilled water, and dried at 50 °C for 24 h and then weighed. The biodegradation based on percent by weight of mass loss was calculated as

$$\text{Biodegradation } (\%) = \frac{W_1 - W_2}{W_1} \times 100 \qquad (2)$$

where $W_1$ is the initial dry weight of the samples, and $W_2$ is the residue dry weight of films of the samples after biodegradation in soil.

Antibacterial tests of NRBC and ANRBC films against *E. coli* and *S. aureus* were determined by the disc diffusion method. *E. coli* or *S. aureus* was suspended in nutrient broth at the standardized concentration of $\approx 10^8$ cells/mL prior to spreading 1 mL the prepared bacterial broth onto an agar plate. Then, the film samples were cut into 1 cm circular discs and were gently placed on the surface of an agar plate. The plates were incubated at 37 °C for 24 h. After that, the clear zones of growth inhibition around sample infused discs were measured.

The method for cytotoxicity tests followed similar procedures to those in previous reports [13,14]. The cytotoxicity was tested against human embryonic kidney cells (HEK 293) and human skin keratinocytes (HaCaT). The cells were cultured in Dulbecco's modified Eagle medium (DMEM) supplemented with 10% fetal bovine serum (FBS), 1% l-glutamine, 100 unit/mL penicillin, 2 mg/L lactalbumin, and 100 µg/mL streptomycin in a humidified incubator with an atmosphere of 5% CO$_2$ at 37 °C. A film sample of 0.5 g was autoclaved at 121 °C for 15 min. The sterilized film was extracted by immersion in 10 mL DMEM at 37 °C for 24 h. Then, the extract was filtrated through a 0.45 µm filter, and the concentration was adjusted to 1000 µg/mL. Cells of $1 \times 10^4$ cells/mL were seeded into a 96-well plate of DMEM at 37 °C and 5% CO$_2$. After 24 h, the DMED was replaced by 100 µL/well of the extract and the cell was further incubated for another 24 h using the same conditions. A positive control was the cell cultured in DMEM. Then MTT (3-(4,5-dimethythiazolyl-2)-2,5-diphenyltetrazolium bromide) assay was carried out to determine the number of living cells. The extract was removed, and then 10 µL of MTT reagent was added into each well. The incubation was performed for 4 h at 37 °C and 5% CO$_2$. After that, MTT solution was removed and 150 µL dimethyl sulfoxide (DMSO) was added into each well to solubilize blue formazan crystals. Colorimetric detection was done by a UV–VIS spectrophotometer (Thermo Scientific, Waltham, MA, USA) at a wavelength of 570 nm. The amount of color produced was directly proportional to the number of viable cells. The viability of cells in the extract medium was compared to that of the control medium (DMED). The percentages of cell viability was expressed as mean ± SD (n = 3).

## 3. Results

### 3.1. Particle Sizes of BC Fibers and NR

Particle size distribution (by number) of the BC fiber analyzed by Zetasizer Instruments is shown in Figure 1, where the average size was 1.249 µm with a polydispersity index (PDI) of 0.910. Rod-shaped fibrils of BC were packed in the form of a small sheet composed of nanofiber networks, in which the diameters of the fibers were around 3–8 nm. The particle size of NR in the NR latex was between 0.01 and 2.0 µm (figure not shown).

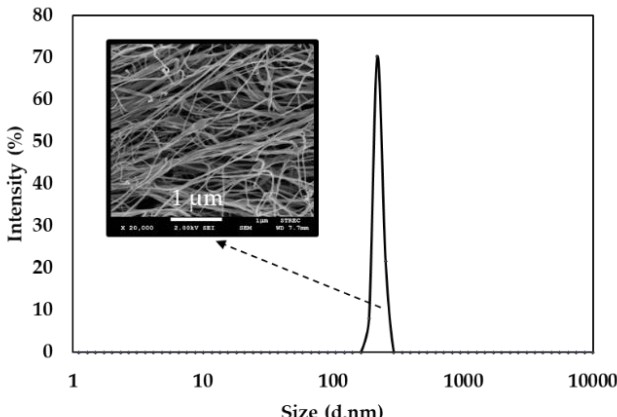

**Figure 1.** Particle size distribution of BC fibers in the slurry.

### 3.2. Scanning Electron Microscopy (SEM)

The outlook and cross-section morphology of NRBC and ANRBC films are shown in Figure 2. The composite films of NRBC and ANRBC had a smooth light-impervious surface. The SEM images of the cross-sectional views showed two immiscible components with good dispersion and good distribution of BC fibers in the NR matrix without fiber agglomeration or precipitation. The layered structures were clearly displayed in NRBC and ANRBC films with ≥50% BC loading. The acid modification showed no significant effect on the film morphology or structure in this observation.

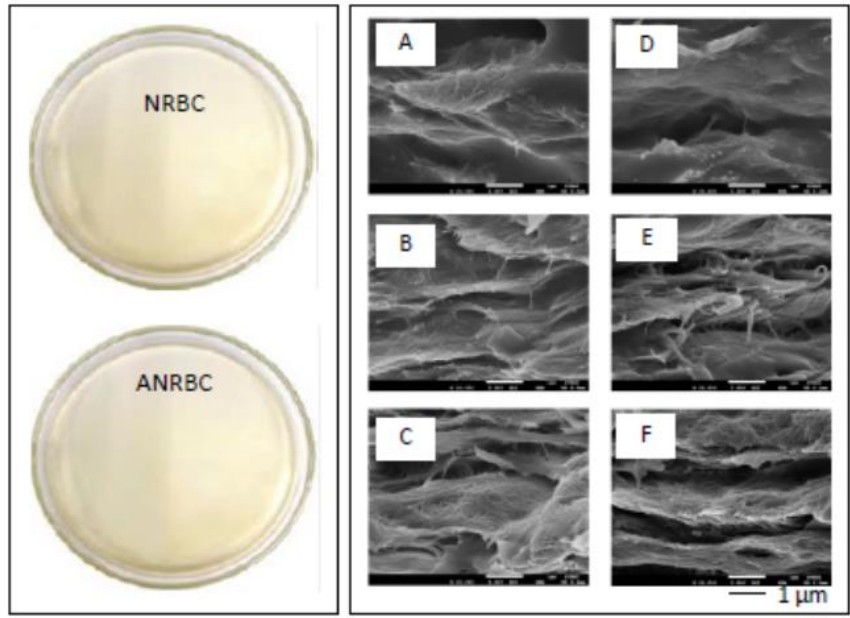

**Figure 2.** The outlook (left) and SEM images of cross section views (right) of NRBC20 (**A**), NRBC50 (**B**), NRBC80 (**C**), ANRBC20 (**D**), ANRBC50 (**E**), and ANRBC80 (**F**).

### 3.3. FTIR Analysis

FTIR spectra of NR, BC, NRBC, and ANRBC composites are shown in Figure 3. NRBC displayed peaks consisting of pure NR and BC. The peaks around 2947 cm$^{-1}$, 2841 cm$^{-1}$, and 1430 cm$^{-1}$ were assigned to-CH$_3$, -CH$_2$, and C=C stretching, respectively, which are the characteristic peaks of NR. The peaks at 3000–3600 cm$^{-1}$ and 996 cm$^{-1}$ were assigned to OH and C–O stretching, respectively, which are the characteristic peaks of BC. The position of characteristic peaks of composites slightly shifted from the peaks of the reactants, which implied the interfacial interaction between the filler and NR matrix. In acid modification, ANRBC composites showed similar patterns and positions of

characteristic peaks, which indicated no chemical bonding interaction after acid modification. However, a slightly shifted position of characteristic peaks was observed, which may imply interfacial interaction at the $CH_2$ position. The modification method was by immersion of dried films of NRBC in 0.25 M lactic acid for 2 h at room temperature and then the films were thoroughly washed with DI water and dried again in an air convection oven. The residual amount of lactic acid in film was very small, and thus it could not be detected from FTIR analysis.

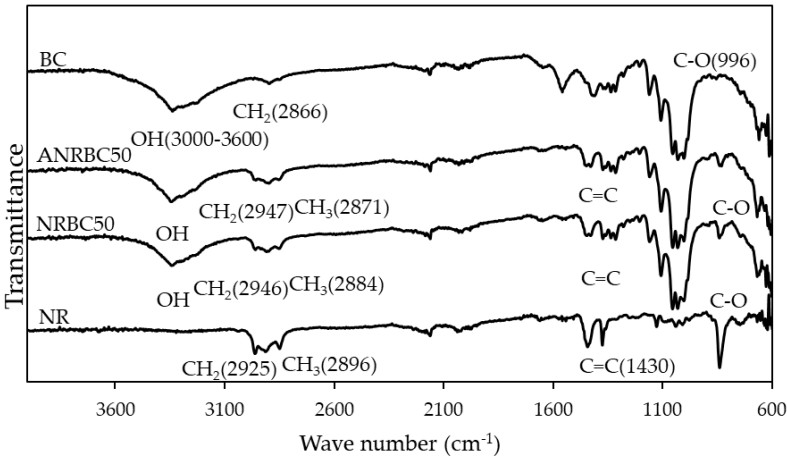

**Figure 3.** FTIR spectra of NR, BC, NRBC, and ANRBC films.

*3.4. Contact Angle Measurement*

The static contact angle was used to identify the improvement of the material hydrophilicity (Figure 4). Because of the hydroxyl groups in cellulose, it has high hydrophilic characteristics and tends to strongly interact with water. BC is considered to be a hydrophilic substance and the degree of water contact angle of BC was found to be around $47.5 \pm 2.8$. However, the degree of water contact angle of NR, which is a hydrophobic polymer, was around $116.2 \pm 7.8$. The dynamic water contact angle of the NRBC composites decreased as BC loading increased. According to lactic acid modification, ANRBC20 showed an increase in hydrophilicity when compared with NRBC20 film owing to a more hydrophilic polymer surface generated by acid treatment. However, there were no significant changes in hydrophilicity between ANRBC and NRBC at BC loadings $\geq 50$ wt.%. Therefore, the acid modification played an important role in hydrophilicity of the NRBC films only for those containing a low content of BC.

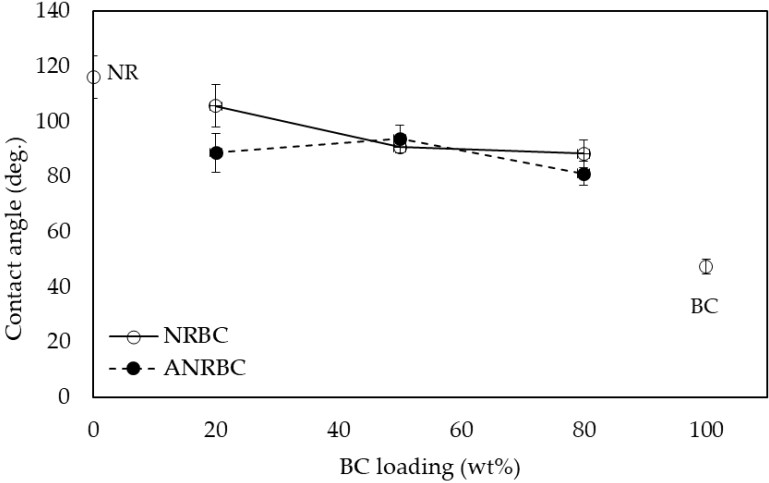

**Figure 4.** The static water contact angle of NR, BC, NRBC, and ANRBC films. The values are expressed as mean $\pm$ SD (n = 3).

### 3.5. XRD Analysis

The XRD pattern and crystallinity of NRBC and ANRBC composite films is shown in Figure 5. The degree of crystallinity of the BC film was 58.2%. As an amorphous polymer, the degree of crystallinity of the NR film was 0. The crystallinity of the composite film was increased with an increase in BC loading. However, the crystallinity of NRBC 80 was lower than that of ANRBC 80. The linear increase in crystallinity of ANRBC with BC content in the composites was observed.

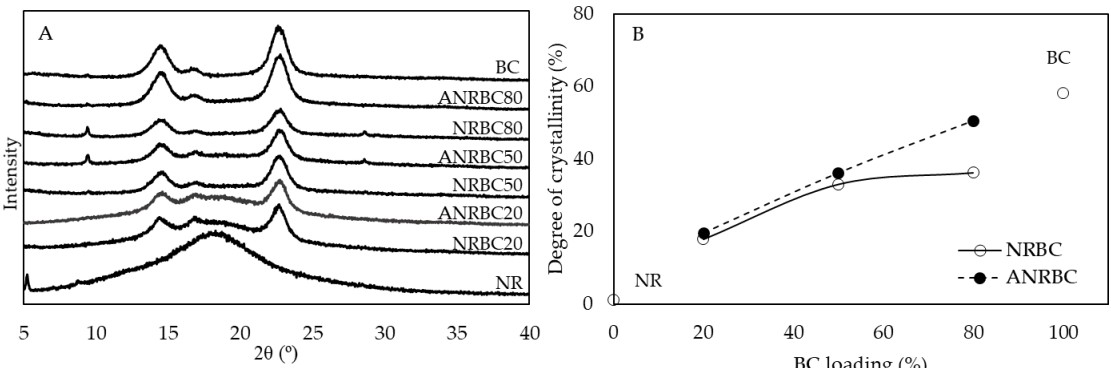

**Figure 5.** XRD pattern (**A**) and degree of crystallinity (**B**) of NR, BC, NRBC, and ANRBC composites.

### 3.6. Thermal Properties

The TGA curves of BC, NR and NRBC, and ANRBC films are shown in Figure 6. For NRBC 20, the decomposition temperatures of polymers were observed to be enhanced compared to pure NR and the other NRBC composites. Similar patterns of TGA curves were observed between NRBC and ANRBC films. The thermal properties of the NR, BC, NRBC, and ANRBC films were investigated in terms of temperatures at maximum weight loss rate ($T_{max}$), melting temperature ($T_m$), and glass transition temperature ($T_g$), as shown in Table 2. It was demonstrated that the NR film presented a higher thermal resistance than BC; the temperatures for the $T_{max}$ of NR and BC were 350 °C and 301 °C, respectively. The $T_{max}$ of the NRBC composites decreased as BC loading increased. Compared with NRBC, ANRBC showed a slight decrease in $T_{max}$. BC showed $T_m$ at 149 °C; however, $T_g$ could not be detected in the test because the $T_g$ of BC may have been below the initial scanning temperature. NR shows $T_m$ and $T_g$ at 267 and −64 °C, respectively. A decrease in $T_m$ of the composites was observed as the BC loading content increased. ANRBC composites showed a relatively higher $T_m$ when compared with the NRBC composites. However, the reinforcement of NR by BC fibers had only a slight effect on $T_g$ of the composites.

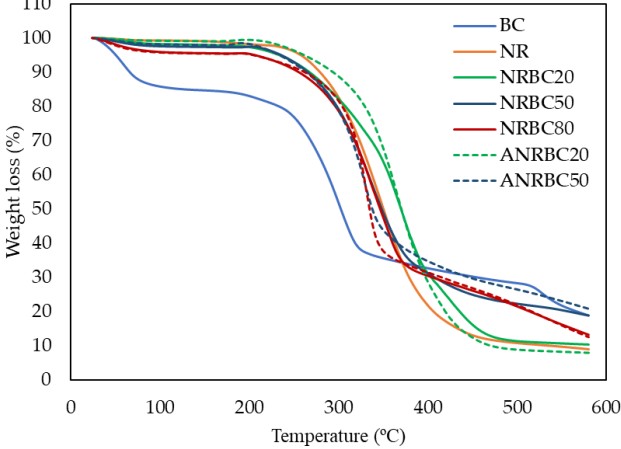

**Figure 6.** Weight loss of NR, BC, and NRBC composites.

**Table 2.** Thermal properties of NR, BC, NRBC, and ANRBC films.

| Samples | $T_{max}$ (°C) | $T_m$ (°C) | $T_g$ (°C) |
|---------|---------|--------|--------|
| NR | 350 | 267 | −64 |
| BC | 301 | 149 | - |
| NRBC20 | 370 | 220 | −63 |
| ANRBC20 | 367 | 280 | −62 |
| NRBC50 | 335 | 176 | −64 |
| ANRBC50 | 329 | 186 | −63 |
| NRBC80 | 334 | 167 | −66 |
| ANRBC80 | 331 | 180 | −65 |

*3.7. Mechanical Properties*

The Young's modulus, tensile strength, and elongation at break of dry films of NR, BC, NRBC, and ANRBC were determined (Figure 7). The values were expressed as mean ±SD from five samples (n = 5) and statistically analyzed by Student's *t*-test for two-sample assuming equal variances in Microsoft office 2010. The differences were considered statistically significance at the level of *p* < 0.05. The Young's modulus and tensile strength of both NRBC and ANRBC films were enhanced with an increase in the BC fiber content. Maximal Young's modulus and tensile strength were obtained with BC loading at 80 wt.%. The lactic acid modification significantly improved Young's modulus (*p* < 0.05) and tensile strength (*p* < 0.05) of ANRBC at 80 wt.% BC loading content as compared to NRBC. The tensile strength of ANRBC80 was approximately 209 times and 1.6 times those of NR and NRBC80 films, respectively. On the other hand, the elongation at break of both NRBC and ANRBC were reduced with an increase in the BC fiber content. Among the NRBC composites, NRBC20 had the highest elongation at break at 15.4%, which was about 0.13 of the NR film. With lactic acid modification, the elongation at break of ANRBC20 was improved to ≈1.3 of NRBC20.

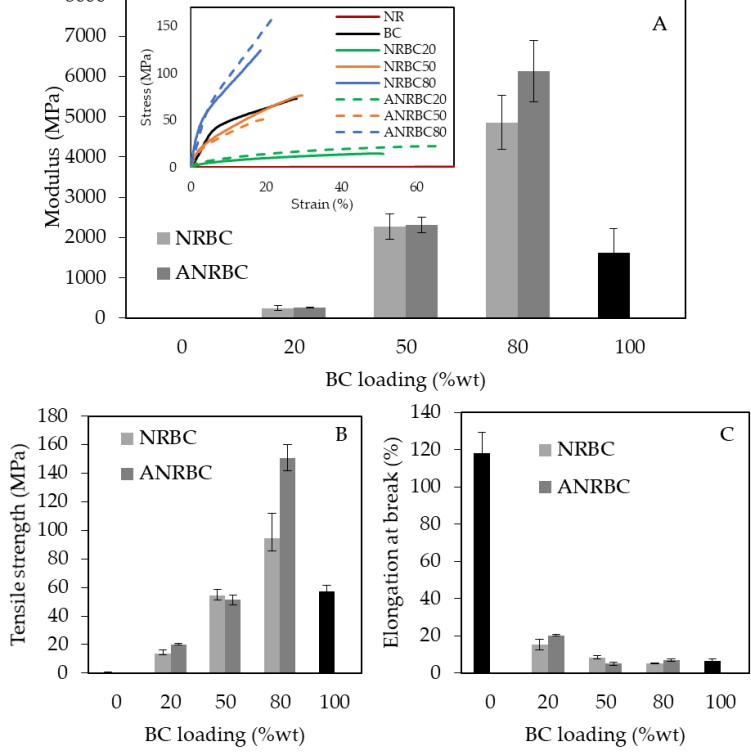

**Figure 7.** Mechanical properties, (**A**) Young's modulus, (**B**) tensile strength and (**C**) elongation at break of NR, BC, NRBC, and ANRBC films. The values are expressed as mean ± SD (n = 5).

### 3.8. Toluene and Water Uptake

The degree of swelling of NRBC and ANRBC films in toluene at equilibrium decreased as BC loading increased, as shown in Figure 8A. The degree of swelling of NR and BC films in toluene were around 2000% and 50%, respectively. Under 20 wt.% BC loading, ANRBC composites swelled in toluene less than did NRBC composites. However, there was no significant improvement in toluene resistance between ANRBC50 and NRBC50 or ANRBC80 and NRBC80. On the other hand, the increase in BC loading in the composite films enhanced the degree of swelling in water, as shown in Figure 8B, whereas the degree of swelling of NR and BC films in water were around 100% and 370%, respectively. Similarly to the swelling of the composite films in toluene, it was found that the acid modification promoted water resistance in the ANRBC films when BC loading was ≥50 wt.%. Overall, ANRBC20 had a low degree of swelling in toluene, which was similar to BC film, and had a low degree of swelling in water, which was similar to NR film.

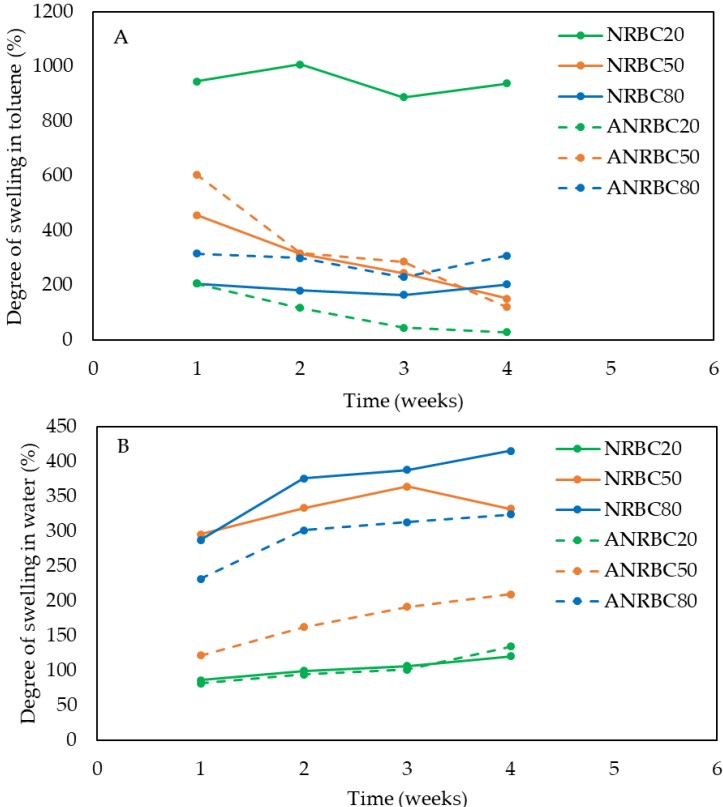

**Figure 8.** Degree of swelling of NRBC and ANRBC films in toluene (**A**) and water (**B**).

### 3.9. WVTR Measurement

As shown in Figure 9, the WVTR of NR was the lowest (≈0), and BC film had the highest WVTR. The WVTR of the composite films was enhanced with increasing BC fiber content. The WVTR of ANRBC at 20 wt.% BC loading was relatively lower than that of NRBC. However, at ≥50 wt.% BC loading, no difference in the WVTR between NRBC and ANRBC was observed.

### 3.10. Biodegradation in Soil

In this work, the biodegradation of NR, NRBC, and ANRBC in soil was determined in the natural soil environment, where the soil moisture was around 11–14 (% vol) and the soil temperature was around 27–33 °C. As shown in Table 3 and Figure 10, BC and the composites of NRBC and ANRBC with 80 wt.% BC loading were completely degraded in soil within 3 months, whereas the weight loss of the pure NR film was about 6% at 3 months. The biodegradability of the composites was increased with

the BC loading content. The acid modification had a slight effect on the reduction of biodegradation of composites with ≤50 wt.% BC loading. However, no reduction effect of biodegradation was observed in ANRBC80 (80 wt.% BC loading) as compared with NRBC80. The biodegradability of NRBC80 and ANRBC80 were comparable to that of BC film.

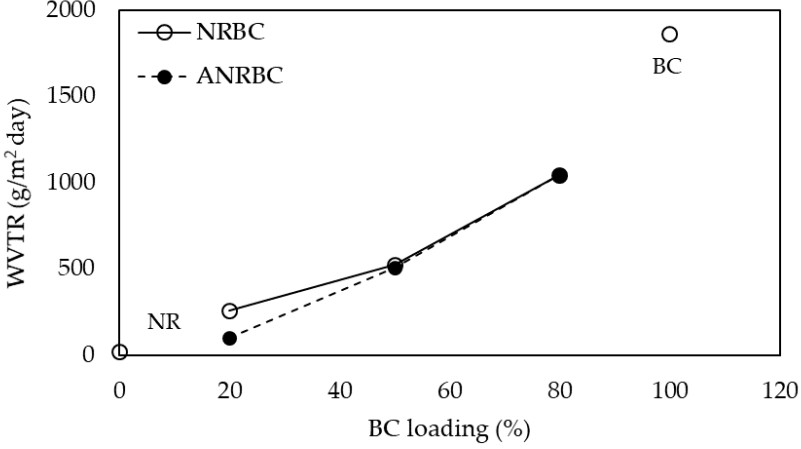

**Figure 9.** Water vapor transmission rates (WVTR) of NR, BC, NRBC, and ANRBC films.

**Table 3.** Weight loss of NR, BC, NRBC, and ANRBC films in soil.

| Samples | Weight Loss (wt.%) | | |
|---|---|---|---|
| | 1 Month | 3 Months | 6 Months |
| NR | 4.08 | 5.95 | 6.83 |
| BC | 34.90 | +++ | +++ |
| NRBC20 | 11.28 | 13.61 | 20.40 |
| ANRBC20 | 9.62 | 12.84 | 15.17 |
| NRBC50 | 10.62 | 21.49 | 44.30 |
| ANRBC50 | 10.53 | 15.46 | 26.53 |
| NRBC80 | 23.91 | +++ | +++ |
| ANRBC80 | 33.28 | +++ | +++ |

+ + + = complete degradation.

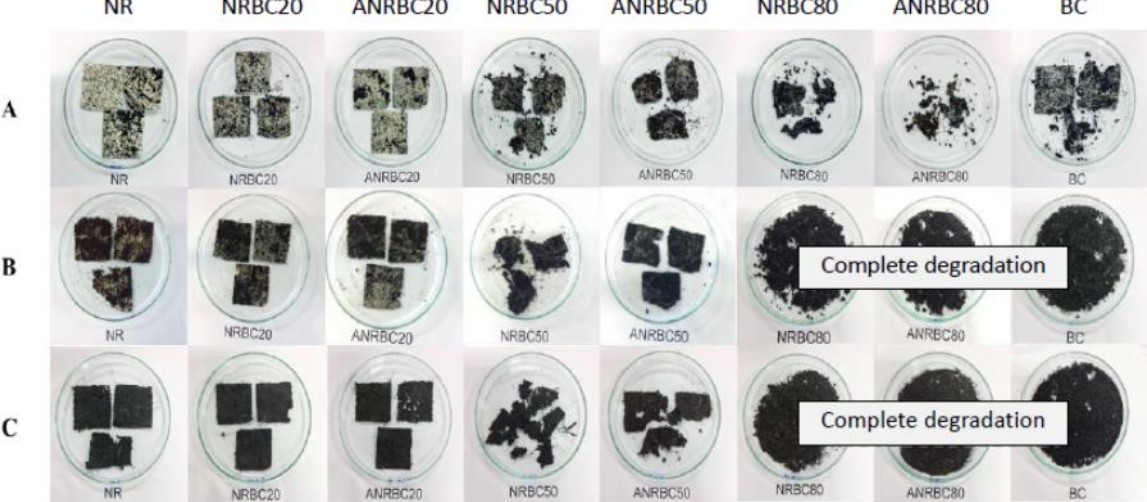

**Figure 10.** Biodegradation in soil of NR, BC, NRBC, and ANRBC films for 1 month (**A**), 3 months (**B**), and 6 months (**C**).

### 3.11. Biological Activities

Since the films might be further used in applications for food packaging and biomaterials, the cytotoxicity of NRBC and ANRBC were therefore evaluated against human embryonic kidney cells (HEK 293) and human skin keratinocytes (HaCaT) by MTT-cell culture assay. Extract medium from the films were used to culture cells, in comparison to a control medium (DMEM). Percentages of cell viability compared to that of the control are shown in Figure 11. The cell viabilities in the extract medium of all films were ≥80%, as compared to the control. Therefore, no films showed toxicity against the HEK 293 and HaCaT cell lines. In addition, because ANRBC films were modified by lactic acid and lactic acid has been widely used to inhibit the growth of important microbial pathogens, such as *E. coli* (Gram-negative bacteria) or *S. aureus* (Gram-positive bacteria), the antibacterial activity of the films was evaluated. The disc diffusion method was used to test the antibacterial activities of the films against *E. coli* and *S. aureus* (figures not shown). Antimicrobial activities were indicated by clear zones of growth inhibition observed around sample infused discs on the agar. The results revealed that no films had a significant impact on the growth of *E. coli or S. aureus*. The result agrees with the observation of no significant effect of the lactic acid modification on biodegradability of ANRBC films as compared to NRBC films.

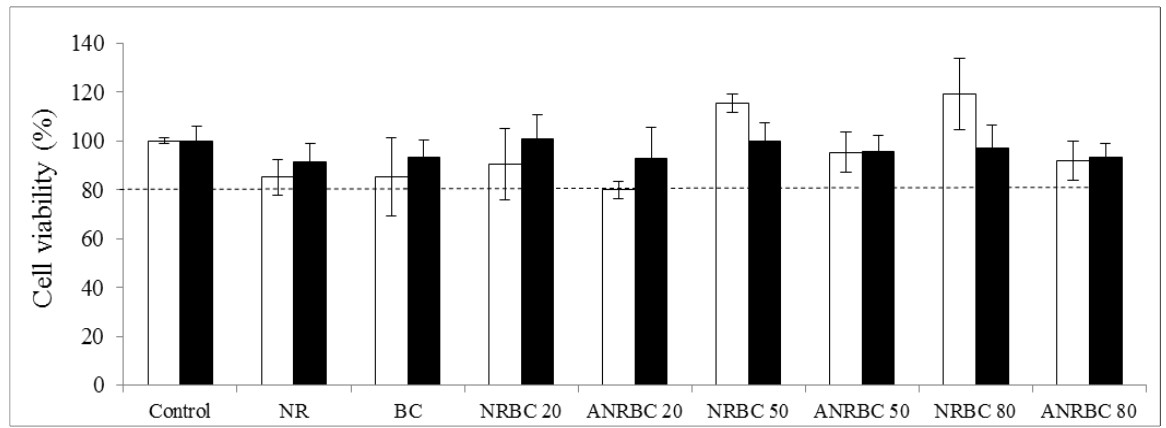

**Figure 11.** Cell viabilities (%) of HEK 293 (□) and HaCaT (■) cultured in extract medium of the NR, BC, NRBC, and ANRBC films for 24 h as compared to the control. the values were expressed as mean ± SD (n = 3).

## 4. Discussion

Films of NRBC were prepared via a latex aqueous microdispersion process and were modified by lactic acid treatment. Significant changes and improvements in some properties were achieved from the acid modification.

The acid modification affected the polarity of films. The polarity could be enhanced with the increase of hydroxyl, carbonyl, and carboxyl groups on the surface of the composite films. In addition, the surface morphology change might also have affected the value of the contact angle. The water contact angle could be decreased because of increasing surface roughness [15]. According to lactic acid modification, ANRBC20 shows a slight increase in hydrophilicity when compared to NRBC20 film due to a more hydrophilic polymer surface generated by acid treatment. According to the FTIR results, it has been suggested that OH groups on the cellulose chain could form a hydrogen bond interaction with lactic acid. The interaction between lactic acid and BC via acid modification could promote the inter- and intra-molecular interaction of OH groups on BC fibers [16]. However, at high BC loadings ≥50 wt.%, there are no significant changes in hydrophilicity between ANRBC and NRBC.

The acid modification showed some effects on the crystallinity of films. Generally, the XRD pattern of BC demonstrated peaks observed at 14.1°, 16.1°, and 22.4° [17,18]. The XRD patterns of the composite films showed three peaks at similar degrees to those of BC. The crystallinity of the composite

films increased with the increase of BC loading content. Significant improvement in crystallinity after lactic modification was observed in ANRBC with 80% BC loading. As compared with NRBC50, the crystallinity of NRBC80 was only slightly increased, which is probably due to the aggregation and disorder distribution of BC fibers when a very high amount of BC was loaded in the NR matrix. However, with lactic acid modification, the presence of the hydrogen bond interaction between the lactic acid and OH groups on the composite surface promoted a uniform distribution of BC fibers in the NR matrix, resulting in higher crystallinity of ANRBC80 as compared with NRBC80. The change in crystallinity of the composite films could imply the reorganization of the film structure.

The acid modification also showed some effects on the thermal properties of the composite films. The temperature for degradation of BC was around 301 °C, which was close to the temperature at which the thermal depolymerization of hemicelluloses and the breakdown of glycosidic linkages of cellulose occurred, as was previously reported for plant celluloses [19,20]. The degradation of the NR film occurred from 220 °C to 450 °C [21]. The $T_{max}$ of the NRBC composites decreased as BC loading increased, owing to the higher thermal stability of the C=C bonds of NR than the glycosidic linkages of BC. Compared to NRBC, ANRBC showed a slight decrease in $T_{max}$, which was possibly due to the inter- and intra-molecular interactions with lactic acid. The improved interfacial interaction between NR and BC and the higher composite homogeneity after the acid modification could promote the heat transfer of the composite films. The disorder or disarrangement of NR crystalline units as the BC fibers integrated into the NR matrix reduced $T_m$ [22–24]. ANRBC composites showed a relatively higher $T_m$ when compared with the NRBC composites. The results supported that the lactic acid modification promoted the homogeneousness between NR and BC, and it may have improved the interfacial interaction between the BC fibers and NR chains, and the intra- and inter-molecular interaction of the BC fibers. Hydrogen bonding led to a decrease in polymer mobility, resulting in increased $T_m$.

Significant improvement in mechanical strength of the composite films was observed by acid modification. In general, pure NR has a low Young's modulus and tensile strength, but a high elongation at break. Conversely, BC has a high Young's modulus and tensile strength, but a low elongation at break. The Young's modulus and tensile strength of both NRBC and ANRBC films were enhanced with an increase in the BC fiber content [4]. Maximal Young's modulus and tensile strength were obtained with BC loading at 80 wt.%. Moreover ANRBC80 showed a higher modulus and tensile strength than those of NRBC80, which was probably due to the OH- inter- and intra-molecular interaction obtained by lactic acid modification. Acid modification increased the rigidity of the composites because the cellulose whiskers with hydrogen bond forming reduced the brittleness of the composite backbone. The enhanced rigidity was possibly due to the increased crystallinity of the ANRBC80, as previously observed from the XRD result (in Section 3.5). ANRBC80 had the highest tensile strength, which was approximately 209 times and 1.6 times that of the NR and NRBC80 films, respectively. In addition, with lactic acid modification, the elongation at break of ANRBC20 was improved to ≈1.3 of NRBC20.

The acid modified films of ANRBC showed significant improvement in solvent resistances. The interaction between the BC fibers and the NR chains in the NR matrix limited the absorption of the non-polar solvent of the composite films. The strong interface between the BC fibers and the NR chains constrained the swelling of the polymeric chains located in the interfacial zone [25,26]. In addition, the BC fibers provided a tortuous path for toluene because of its polarity difference. The decrease in solvent uptake in the composites was probably due to the increase in the tortuous path and the reduction in the transport area [27]. Furthermore, the ability of the composites to swell in toluene was also due to the behavior of the filler itself. BC and most of its derivatives were insoluble in non-polar solvents [28]; therefore, the addition of BC improved the non-polar resistance of the composites. Because of acid modification, ANRBC composite films had less swelling and were not dissolved in either the non-polar or polar solvents. Under 20 wt.% BC loading, ANRBC composites swelled in toluene less than did the NRBC composites. Acid modification partially restricted the swelling of the composites in non-polar solvent. The hydrogen bond probably induced intra- and inter-species interactions within

the composite films. Structure and molecular packing behavior of the composites with high molecular packing density swelled less than those with loose molecular packing. Furthermore, hydrogen bond formation improved the interfacial interactions and subsequently restricted the diffusion of toluene molecules in the vicinity of the cellulosic surface. However, there was no significant improvement in toluene resistance between ANRBC50 and NRBC50 or ANRBC80 and NRBC80. Therefore, at high BC loading, the effect of acid modification might be inferior to the effect of interfacial polar and non-polar interactions between cellulose and toluene. On the other hand, the increase in BC loading in the composite films enhanced the degree of swelling in water, which was due to the hydrophilicity of the BC fibers. This can be attributed to the increasing presence of OH groups in the composite films. However, the hydrophilization of the NR chains increased their sensitivity to water [22]. In addition, the presence of natural fibers in the compound disturbs the structural homogeneity in the material, which produces voids at the interface and may increase the ability of water molecules to penetrate the composite through capillary transport. The compatibility between the phases should lead to a decrease of voids in the composites [29]. Similarly to the swelling of the composite films in toluene, the improved molecular packing and interfacial interactions between the BC fiber and NR chains by lactic acid modification reduced the degree of swelling in water. The acid modification promoted water resistance in the ANRBC films when BC loading was ≥ 50 wt.%. The improved NR–BC interfacial interaction and their compatibility should decrease void volume in the composites and, accordingly, limit water absorption in the composite films. Among all the composite films, ANRBC20 had the highest structural stability against a non-polar solvent (toluene) and water; consequently, its degrees of swelling in toluene and in water were the lowest among the composites.

The WVTR is an important property for packaging applications. Generally, WVTR varies in factors such as the viscosity of forming solution, film formation procedure, composition, and film thickness [30,31]. The WVTR of the composite films was enhanced with increasing BC fiber content. The increase in the WVTR is possibly based on the increase in the degree swelling in water and the hydrophilicity of the films because of the high hydrophilic nature of the BC fibers. Previously, poor interfacial attraction between cellulose fibers in agar-based bionanocomposites increased the diffusion of water vapor through the void between the fibers and agar [32]. The crystallinity and the rod shape of BC whiskers also affected the mechanism of water transmission [33]. The WVTR of ANRBC at 20 wt.% BC loading was relatively lower than that of NRBC, which can be explained by the physicochemical characteristics of the acid modified surfaces and the effect of hydrogen bond formation leading to decreased porosity. However, at ≥50 wt.% BC loading, no difference in the WVTR between NRBC and ANRBC was observed because acid modification might have only a minor effect on the WVTR as compared with the effect of the hydrophilicity of composites with a high BC concentration.

Biodegradable properties are an important requirement for green packaging materials. Many studies have been carried out on the degradation of rubber [34]. NR slowly degrades in nature because the specific bacteria that use rubber as a sole carbon source grow quite slowly [33]. Many bacterial strains that use rubber as the sole source of carbon and energy have been previously reported [35–37]. Thus, the biodegradability of the composites was increased with the BC loading content because of the much higher biodegradation rate of cellulose in soil as compared with that of NR [38]. Nanocellulose fibers in the composite films were consumed by soil microorganisms much faster than was NR, which led to increased porosity, void formation, and a loss of integrity of the NR matrix. The rubber matrix was thus broken down into smaller particles. Previously, disintegration of nanocomposite films containing cellulose whiskers was reported to be faster than that of pure NR films [33]. Previously, the average degradation rate of NR/cellulose composites with 10 wt.% cellulose loading was reported, and when cross linked with KOH solution, the average degradation was approximately 0.15 times that of the non-cross linked counterparts [39]. However, in this study, the acid modification had only a slight effect on the reduction of the biodegradation of composites with ≤50 wt.% BC loading, and there was no significant reduction effect of the acid modification on the biodegradation of ANRBC80 (80 wt.% BC loading) as compared with NRBC80. NRBC and

ANRBC films did not show toxicity against the HEK 293 or HaCaT cell lines. Furthermore, by the disc diffusion method, no antibacterial effect of NRBC and ANRBC against *E. coli* and *S. aureus* was detected, which implied no significant release of lactic acid from the acid modification into the agar. ANRBC composites showed significant improvement in mechanical properties, melting temperature, and high resistance to polar and non-polar solvents. The material itself is biodegradable and non-toxic to cells, making it an ideal material in applications for food packaging and biomaterials.

## 5. Conclusions

The interfacial interaction between NR and BC in NRBC composite films and the structural stability were improved by lactic acid modification. The hydrogen bonding that occurred via the acid modification promoted the inter- and intra-molecular interaction of OH groups in the composite films, resulting in an enhanced compatibility of NR chains and BC fibers, which consequently improved thermal stability, solvent resistance, and mechanical strength. No significant effect of the lactic acid modification on film biodegradability in soil was observed; NRBC and ANRBC composites at ≥80 wt.% BC loading were completely degraded in soil within 3 months. According to the in vitro cytotoxicity testing against HEK 293 or HaCaT cell lines, the cell viabilities in the extract medium of all films were ≥80%, as compared to the control. Therefore, the results are encouraging, but additional experiments are required to confirm lack of cytotoxicity. Therefore, ANRBC composites could potentially be used as alternative green materials for disposable products and packaging. In addition, the materials could also be further developed for biomedical applications.

**Author Contributions:** Conceptualization, M.P. and S.P.; methodology, M.P. and S.P.; validation, M.P. and S.P.; formal analysis, S.P.; investigation, M.P. and S.P.; resources, M.P.; data curation, S.P.; writing—original draft preparation, S.P.; writing—review and editing, M.P.; supervision, M.P.; project administration, M.P.; funding acquisition, M.P. and S.P. All authors have read and agreed to the published version of the manuscript.

**Funding:** This research was funded by Thailand research fund (TRF), grant number RGJ-PHD-19.

**Acknowledgments:** The authors acknowledge the support from Chulalongkorn University and the Thailand research fund (TRF) under the Royal Golden Jubilee Ph.D. (RGJ-PHD) Program and TRF (RGU62).

**Conflicts of Interest:** The authors declare no conflict of interest.

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
