# Peer review of "Lactic Acid Modified Natural Rubber–Bacterial Cellulose Composites"

_applsci, doi:10.3390/app10103583_

Round 1

Reviewer 1 Report

In this manuscript, the author designed a lactic acid modified natural rubber-bacterial cellulose composites that showed significant improvement in mechanical properties, melting temperature and high resistance to polar and non-polar solvents. The material itself is biodegradable and non-toxic to cels, making it an ideal materials in the application for packaging and biomaterials. Although the authors did a very comprehensive characterization to the new platforms, there are two small questions needs to be solved. 

  1. the characterization of lactic acid modification is kind of weak. In Page 4, Line 163-164. the lactic acid modification should give C=O stretching signal at 1680-1750 nm-1. However, we didn't observe that peak from ANRBC. Please explain. 
  2. Although the author claimed there is no cytotoxicity, but no data was shown. It's better to describe how you perform the assay and show some representative results

Reviewer 3 Report

Summary: the paper describes the fabrication of a hybrid bacterial cellulose/natural rubber hybrid material which shows improved mechanical properties over pure bacterial cellulose or natural rubber materials as well as improved resistance to chemical degradation. The paper is straightforward, the results are clearly presented and the potential for the development of new green composite materials great. Overall it is a sound paper.

The paper could be strengthened by the following minor additions:
The authors thoroughly characterize the materials used in the fabrication of this composite, but do not characterize the raw materials in the state that they used to fabricate the materials – a mention of the size of the latex polymers and the average length/size of the bacteria cellulose fragments prior the compositing process would help others repeat these materials and also provide another dimension for material tuning.
The authors test the biocompatibility of the material for both bacterial and mammalian cell – this is a good addition to the paper – but I would like more of a discussion of the rational why they did this and how their results are important for application. For instance, why is the lack of an antibacterial property of a material import. I suspect that this property is important for biodegradability, but I think the authors need to explain this more clearly.

Minor wording in introduction
Line 24: “However, there are some defects, such as low strength and abrasion resistance.” These properties are not defects, but rather less desirable….may “However, NR has some less desirable properties such as…….”
Line 31: “BC is produced by A. xylinium in form of microfibrils.” BC is generated as a nanofiber, not microfiber

Line 55-58: “ The research aimed to develop biodegradable films of NRBC composites with improved mechanical properties, good resistance in water and non-polar solvents together with high thermal stability and non-toxicity in order to be further applied in the fields of packaging and biomaterials.”
Better wording: “The goal of this research presented in this manuscript is to develop….”

Round 2

Reviewer 2 Report

Thank you for providing more information regarding the methodology.

The described cytotoxicity tests are extract tests conducted in media for 24 hours.

What was the temperature?

Was any color change of the media observed?

What was area of samples for cytotoxicity tests—presumably the thickness was the same as in Table 1. The area of sample to media volume ratio is *crucial* for assessing extraction, in accordance with ISO10559—especially for such thin films.

What does it mean that the concentration was adjusted to 1 mg/mL? Does this mean the mass of sample to media volume? If so, then this is far too low to be sure of adequate sensitivity. Was a positive control used, to ensure the conditions were sensitive enough to detect toxicity?

If not, I recommend softening the language, because the data cannot fully support the claim of "no cytotoxicity". The results are encouraging, but additional experiments are required to confirm lack of cytotoxicity.
